# Survival analysis of parking duration in urban commercial areas: A case study in Zhengzhou, China

Zhendong Sun[1], Jiangling Wu[2]*, Feng Wang[1], Zhenzhong Tian[1], Qiang He[2]

1 Intelligent Traffic Police Affairs Laboratory, Henan Police College, Zhengzhou, China, 2 School of Civil Engineering and Architecture, Henan University, Kaifeng, China

* wujiangling_2006@126.com

## Abstract

This paper studies the parking demand characteristics of large commercial areas in the city's central regions. The study uses non-parametric and semi-parametric analysis methods in survival analysis to explore if and how weather conditions, parking tariffs, and temporal factors (weekdays, weekends, and short holidays) impact the parking duration. The parking data of a large commercial supermarket in Zhengzhou was collected over one month. Single-factor analysis based on the Product-Limit (PL) approach suggests that the cumulative survival and relative risk curves of parking duration exhibit slight variations across different temporal categories and weather conditions. Based on Cox semi-parametric multi-factor analysis results, the parking duration is significantly influenced by weekdays (regression coefficient = 0.068, hazard ratio = 1.071, P < 0.001), weekends (regression coefficient = 0.042, hazard ratio = 1.043, P < 0.001), moderate rain (regression coefficient = -0.089, hazard ratio = 0.914, P < 0.001), and heavy rain (regression coefficient = 0.030, hazard ratio = 1.030, P = 0.034 < 0.05). The results have indicated that within the study area, compared to short holidays, the parking duration on weekdays and weekends is shorter, with the probability of vehicles ending their parking increased by 7.1% and 4.3%, respectively. Under different weather conditions, compared to sunny days, parking duration is longer during moderate rain, with the probability of vehicles departing decreased by 8.6%, whereas during heavy rain, parking duration is shorter, with the probability of vehicles departing increased by 3%. Notably, the parking tariffs demonstrated no statistically significant impact. These findings suggest that temporal variations and rainfall patterns should inform dynamic parking management strategies, including weather-responsive pricing adjustments and spatial capacity optimization during peak periods.

**Data availability statement:** The raw data contains sensitive information (e.g., vehicle license plates) and cannot be shared publicly due to privacy restrictions. However, a fully de-identified dataset is provided as supplementary material with this submission. This dataset retains aggregated parking duration metrics and anonymized vehicle identifiers (license plates replaced with SHA-256 hashes). Importantly, these anonymization procedures do not affect the computational processes or the validity of the research results.

**Funding:** This research was supported by the Science and Technology Project of Henan Province (No. 232102240019). As per our funding disclosure, "The funders had no role in study design, data collection and analysis, decision to publish, or preparation of the manuscript."

**Competing interests:** The authors have declared that no competing interests exist.

# 1 Introduction

The rapid economic development has led to a significant increase in the number of motor vehicles in China, with a growing demand for motor vehicle travel. According to the Ministry of Public Security statistics, by 2022, the number of motor vehicles nationwide has reached 417 million, with 502 million licensed drivers. At the same time, the nationwide parking spaces shortage amounted to 80 million, making parking difficulties a prominent issue. Taking Zhengzhou as an example, as one of the national regional central cities, its motor vehicle ownership ranks fifth in China. According to data provided by the Traffic Police Detachment of Zhengzhou Public Security Bureau, by the end of March 2022, the number of motor vehicles in Zhengzhou's urban area reached 5.12 million. Correspondingly, information from the Henan Provincial portal revealed that by the end of August 2022, Zhengzhou had a parking space shortage of about 1.33 million, far below the national requirement of a 1.1 to 1.3 ratio of motor vehicles to parking spaces. The parking issue is particularly acute. In response to the contradiction between the severe shortage of parking spaces and the increasing demand for parking, the State Council issued the "14th Five-Year Plan for the Development of a Modern Comprehensive Transportation System" in 2021. The plan showed the need to "optimize the supply of parking facilities by category and zone, and improve the utilization efficiency and refined management of parking resources." Characteristic indicators of parking demand, such as parking space turnover rate and parking duration characteristics, are important bases for refined management of parking services.

The current research on parking characteristics mainly focuses on identifying influencing factors and predicting parking demand [1–6]. In terms of the research on parking characteristic indicators, Li, et al. [7] analyzed parking characteristic data of commercial parking lots and tertiary general hospital parking lots using synchronous statistical inference methods and found that parking demand, duration distribution, and departure characteristics all exhibit relative stability, indicating that the distribution characteristics of parking duration have statistical significance for analysis. Regarding the research on parking duration characteristics, Jelena Simićević et al. [8] constructed a logistic model to analyze the impact of parking fees and duration on vehicle usage and parking behavior. Their results indicated that fees primarily influence vehicle usage, while parking time restrictions more significantly affect the choice of parking type. Taisir Khedaywi et al. [9] constructed a multinomial Logit (MNL) model to analyze the preference characteristics of parking space choices among residents in the CBD area of Jordan City. The results showed that the probability of choosing "off-street parking" increases with the duration of parking, and parking demand is positively correlated with both the distance from the destination to the parking lot and the duration of parking. Sana Ben Hassine et al. [10] also constructed an MNL (Multinomial Logit) model to study the parking choice preferences of residents in the city center of Tunisia. The results showed that parking fees, the time spent searching for a parking spot, and walking time all had significant impacts on parking choices. Xiaofei Ye et al. [11] constructed an MNL (Multinomial Logit) model tailored to the operational mode of shared parking lots to investigate the

preference characteristics of residents when choosing parking apps. The results indicated that, given advance notice of parking information to travelers, the availability of parking spaces is the most significant factor influencing the choice of parking apps, while parking fees and distance to the destination are secondary factors. Wei Jia Hong et al. [12] designed a dynamic shared parking method based on a genetic algorithm. Pengfei Zhao et al. [13] took Beijing as the research object to explore the relationship between shared parking and environmental benefits. Janak Parmar et al. [14] developed an artificial neural network to analyze the relationship between parking duration and driver characteristics in Indian cities, concluding that parking fees significantly impact parking duration. In the research on the formulation of parking fee schemes, E.R.Magsino et al. [15] evaluated two time-based dynamic parking pricing methods along with three spatiotemporal dynamic parking pricing methods, encompassing fixed-rate pricing, linear rate pricing, min-max rate pricing, adaptive rate pricing, and complementary rate pricing. The findings reveal that parking fees determined by time value are heavily reliant on the duration of parking by users, whereas those determined by space value consider both the number of available parking spaces upon vehicle entry and the land valuation. In some other related studies. Joshua Schmid et al. [16] investigated the parking duration characteristics of commercial vehicles in New York City, USA, finding that the type of delivered goods and whether parking was illegal influence parking duration. Kaleb Phipps et al. [17] quantified the uncertainties in parking duration for electric vehicles (while charging), improving the management efficiency of parking for charging vehicles. A. Dupont et al. [18] combined the uncertainties in parking duration with factors such as the charging demands of electric vehicles, parking lot capacity, and grid power supply capability to construct a predictive model for parking demand and charging load, and then the optimal pricing strategy was obtained through optimization algorithms. Biruk Gebremedhin Mesfin et al. [19] used a system dynamics model to compare the impacts of parking policies in Shanghai and Zurich on traffic networks with different infrastructure, socioeconomic, and policy characteristics. Most of the studies in the literature focused on parking duration and commercial or electric vehicles. However, there is a lack of research explicitly analyzing the parking duration characteristics of parking lots attached to shopping malls in urban center areas, particularly for leisure and recreation.

Urban central areas are characterized by high levels of economic development, high population density, intense land resource competition, and high demand for underground parking garage services [20].In the central business district of China 's cities, it is a comprehensive multi-functional area composed of high-rise buildings such as commercial buildings, office buildings and hotels, which gathers business, finance, culture, leisure and entertainment, catering and other different types of settings. It has strong traffic attractions, and parking issues often extend to the surrounding areas. Thus, more refined parking service management is required. However, little research has explicitly analyzed the characteristics of parking duration in urban central commercial areas. This research uses survival analysis to study the characteristics of parking duration in urban central commercial areas, uncovering significant factors that influence parking duration and providing theoretical support for refined parking service management. The major contributions of this paper are as follows:

1) This study obtained 34 days of parking data from an underground parking lot at a major shopping center in the central district of Zhengzhou, which provides a solid foundation for further research on parking demand characteristics.

2) Used the product-limit (PL) method to perform univariate analysis on the characteristics of parking duration, and the cumulative survival curves of parking duration under different weather conditions and date types are obtained.

3) Based on Cox semi-parametric multi-factor analysis, we found that the probability of longer vehicle duration is higher during heavy rain, while parking fees showed no significant impact on parking duration.

The structure of this paper is as follows: In section 1, we introduced the concept and principles of survival analysis and analyzed the feasibility of its application in studying the characteristics of parking duration. In section 2, we constructed a PL non-parametric survival estimation model, and a Cox semi-parametric survival analysis model based on univariate and multivariate analyses, respectively. In section 3, we preprocessed the original parking data obtained and provided a

descriptive analysis of the study area. In section 4, we input the processed data into the survival analysis models constructed in the second part for computation and discussed the results. In section 5, we summarized the entire experimental results and proposed corresponding suggestions for parking management based on the conclusions.

## 2  Survival analysis methods and applications

Survival analysis is a method for studying survival phenomena and response time data. It primarily investigates the relationship between the duration of events from the initiation to the termination and the relationships and impacts of related influencing factors. As a longstanding and well-established statistical approach, survival analysis has been widely applied in biology, medicine, and reliability engineering [21]. Its application is particularly notable in medicine, where it is often used to study the survival characteristics of cancer patients or other patients under different treatment regimens [22–23]. Survival time refers to the duration from the start of an initial event to the occurrence of a targeted endpoint event, also known as failure time. For instance, it can refer to the time from implementing different treatment regimens to the onset of illness or death in cancer patients. Survival data consists of two types: complete data and censored data. Complete data, also known as terminal data or complete lifetime data, refers to cases where the survival time of the subject occurs within the observation window, providing complete temporal information. Censored data refers to instances where the event of interest (such as illness or death) does not occur within the observation window for some study subjects or where survival time cannot be observed due to reasons such as loss of follow-up or withdrawal from the study. Censoring includes left censoring, interval censoring, and right censoring.

Some scholars have recently introduced survival analysis methods in transportation engineering research. For example, Kieran Kalair et al. [24] employed a method that combined multiple models, including the Cox regression model, to predict the duration of traffic incidents on the M25 London Orbital motorway, significantly enhancing the interpretability of machine learning models. Yating Zhu et al. [25] constructed a Cox semi-parametric model to analyze the cruising time data of vehicles searching for parking spaces in urban CBD areas, assessing the distribution characteristics of cruising time under different conditions. Xiaofeng Ji et al. [26] used the proportional hazards regression model to analyze overtaking times for vehicles on mountainous two-lane roads. Jiangling Wu et al. [27] constructed a Cox semi-parametric regression model to analyze the duration of lane changes on highways. Fu Guo et al. [28] used survival analysis methods to analyze pedestrian crossing decision times under different conditions. Rui Sun et al. [29] applied survival analysis to investigate the characteristics of bus travel times between stops. Linbo Li et al. [30] used a Cox regression model to predict nighttime parking demand with notable results. Ning Huajing et al. [31] adopted survival analysis methods to analyze 223 segments of urban expressway crash video data from 2016 to 2020 in Hefei, and studied the impact of 14 unsafe behaviors on the duration of traffic accidents. The results showed that for sideswipe collisions, behaviors such as continuously changing multiple lanes, unsafe merging, illegal lane changing, and lane changing without checking the rearview mirror significantly increased the risk of accidents; for rear-end collisions, behaviors such as improper parking, conflict driving, and distracted driving significantly increased the risk of accidents. Meanwhile, the Product-Limit method was used to estimate the survival probability curves of sideswipe and rear-end collisions, and it can be seen from the curves that there are significant differences in the duration of these two types of traffic accidents, which require separate handling. Roxan Saleh et al. [32] were the first to apply survival analysis to investigate the median lifespan of traffic signs in different geographic locations in Croatia and Sweden, finding that blue signs had the longest lifespan, while white signs in Sweden and red signs in Croatia had the shortest. Sagar Keshari et al. [33] recorded a total of 250 overtaking maneuvers by motorcycles and mopeds and constructed a hazard-based duration model to analyze the factors influencing overtaking duration, finding that initial gap, final gap, speed of the overtaken vehicle, and multiple overtakes all had significant positive impacts on overtaking duration. In summary, survival analysis can be used in traffic engineering to study the characteristics and factors influencing time duration.

## 3 Model construction

### 3.1 Survival function of parking duration.

When analyzing parking duration, the survival function is a fundamental function used to describe the statistical characteristics of parking duration. It is the probability that the parking duration of an individual exceeds time $t$, the probability that the vehicle survives until time $t$ and leaves after time $t$. The hazard function is another basic function in survival analysis, also known as the failure function. It describes the probability that an individual, given its survival up to a certain time, will experience the event of interest (in this case, leaving the parking lot) with the subsequent interval $\Delta t$.

Let $T$ represent the parking duration of the target vehicle, with $T \geq 0$, then the cumulative distribution function is given by formula (1).

$$F(t) = P(T \leq t)$$
$$= \int_0^t f(x)dx, \forall t \geq 0 \tag{1}$$

where $P(T \leq t)$ denotes the probability that the event of completing parking and leaving the lot occurs by time $t$, and $f(x)$ is the probability density function of $T$, expressed as formula (2).

$$f(t) = \frac{dF(t)}{dt}$$
$$= \lim_{\Delta t \to 0} \frac{P(t \leq T \leq t+\Delta t)}{\Delta t}, \forall t \geq 0 \tag{2}$$

The survival function is the probability that the parking duration of the target vehicle is greater than $t$ when it stops, denoted by $S(t)$ and is given by formula (3).

$$S(t) = P(T > t)$$
$$= 1 - F(t)$$
$$= \int_t^\infty f(x)dx, \forall t \geq 0 \tag{3}$$

Let $h(t)$ denote the hazard function corresponding to the survival function. It represents the probability that a vehicle parked up to time $t$ will leave the parking lot within the next infinitesimal time interval $\Delta t$. The mathematical expression for the hazard function is as formula (4).

$$h(t) = \lim_{\Delta t \to 0} \frac{P(t < T < t+\Delta t | T \geq t)}{\Delta t}$$
$$= \lim_{\Delta t \to 0} \frac{S(t) - S(t+\Delta t)}{\Delta t S(t)}, \forall t \geq 0 \tag{4}$$

### 3.2 Product-Limit non-parametric survival model.

The product limit (PL) estimate was first proposed by Kaplan and Meier, also known as the Kaplan-Meier (K-M) estimate [34]. This estimation model requires that the survival time data and right-censored data are fully known. Survival times of $n$ individuals are observed for a sample size of $N$, resulting in a dataset comprising both terminal and censored data, denoted as $t_1, t_2, \ldots t_n$. When $t_i$ represents a terminal event, let $\delta_i = 1$; if it represents right-censored data, then $\delta_i = 0$. The data is arranged in ascending order (with terminal data preceding censored data when they are equal) as follows: $t(1) \leq t(2) \leq \ldots t(n)$. Thus, the PL estimate of $S(t)$ can be defined as formula (5).

$$\hat{S}(t) = \begin{cases} 1, & t \in [0, t_{(1)}), \\ \prod_{i=1}^{j} \left(\frac{n-i}{n-i+1}\right)^{\delta_{(i)}}, & t \in [t_{(j)}, t_{(j+1)}), j = 1, 2, \ldots, n-1, \\ 0, & t \in [t_{(n)}, \infty). \end{cases} \tag{5}$$

## 3.3 Cox regression analysis model

The Cox regression analysis model, also known as the proportional hazards model, uses survival time and event status as dependent variables to analyze the effects of different independent variables on the dependent variable. Let $X = (X_1, X_2, ..., X_k)$ be the $k$ covariates that affect the parking duration of a vehicle. Then $h(t, x)$ is the hazard function at time $t$ under the influence of factor $X$, and its mathematical expression is as formula (6).

$$h(t, X) = h_0(t) \cdot e^{\beta X} = h_0(t)e^{(\beta_1 X_1 + \beta_2 X_2 + ... \beta_k X_k)} \tag{6}$$

In equation (6), $\beta = (\beta_1, \beta_2, ..., \beta_k)$ is the partial regression coefficient, and $h_0(t)$ is the baseline hazard function, which represents the hazard function at time $t$ when $X_1 = X_2 = ... = X_k = 0$.

# 4 Data pre-processing

## 4.1 Overview of the study area

This study examines the parking data from an underground parking lot at a major shopping center in Zhengzhou's central district from April 28 to May 31, 2023. The study area is shown in Figure 1. The shopping center covers a total area of 54 acres, with a building area of approximately 400,000 square meters, and is equipped with 1,272 underground parking spaces. There are no ground-level parking facilities. The shopping center operates from 10:00 AM to 10:00 PM. The parking lot operates on a cumulative fee-based model: parking is free for the first 20 minutes, 5 yuan for parking between 20 and 60 minutes, and an additional 2 yuan for every subsequent hour or part thereof. The specific calculation can be represented by formula (7).

$$C(t) = \begin{cases} 0, & 0 \le t \le 20 \\ 5, & 20 < t \le 60 \\ 5 + 2(\lfloor \frac{t-60}{60} \rfloor + \max(0, \min(1, \frac{t-60}{60} - \lfloor \frac{t-60}{60} \rfloor))), & t > 60 \end{cases} \tag{7}$$

Where $C(t)$ represents the parking fee in yuan (CNY), and $t$ represents the parking duration in minutes.

Data collection over the 34 days resulted in 188,047 entries from April 28 to May 31, including variables such as license plate number, entry and exit dates and time, and vehicle type. The entire 24-hour day, starting from 00:00, is divided into hourly intervals from 0 to 23. A comprehensive analysis of these parking data reveals distinct temporal characteristics in parking demand. The highest demand was observed during the Labor Day holiday, followed by weekends, as shown in Figs 2–4.

Figs 2 to 4 show that the peak parking times for the target parking lot on holidays (including short holidays and regular weekends) are between 11:00 AM and 12:00 PM and between 4:00 PM and 5:00 PM. During non-holidays, the peak parking times are between 12:00 PM and 1:00 PM and between 5:00 PM and 6:00 PM. There is a noticeable shift in peak holiday demand times compared to non-holidays. To analyze the utilization of this parking lot, indicators such as parking space turnover rate and peak parking saturation are used for evaluation. The parking space turnover rate refers to the number of times a single parking space is occupied within a unit time period. This indicator directly reflects the utilization efficiency of the parking spaces. The higher the turnover rate, the more frequent the parking and departing, indicating higher service efficiency of the parking spaces. It can be calculated using the following formula (8),

$$R = \frac{c_t}{W} \tag{8}$$

Where $R$ represents the parking space turnover rate, $ct$ is the number of parking within a unit time period, and $W$ is the total number of parking spaces. The calculation results are shown in Table 1.

 

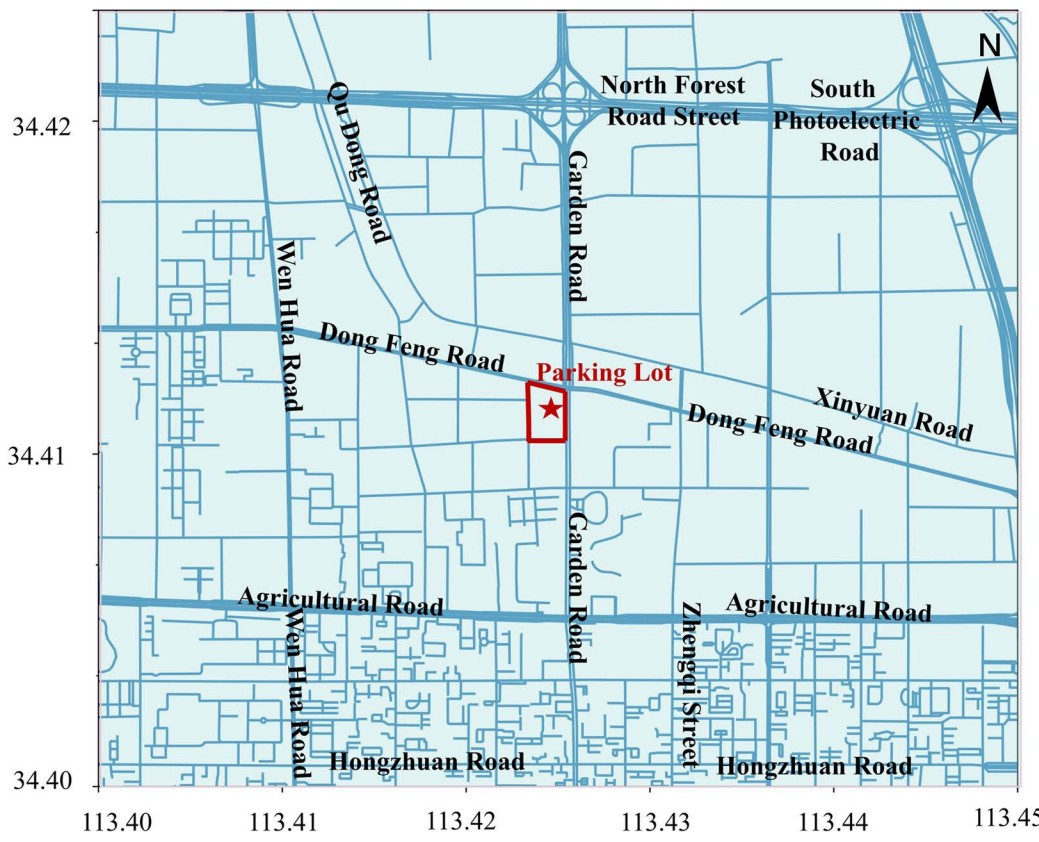

**Fig 1. The diagram of study area.** Base map data © OpenStreetMap contributors (Open Database License, ODbL; https://opendatacommons. org/licenses/odbl/). This cartography is a derivative work of OpenStreetMap data under the ODbL. Commercial area boundaries and parking locations were annotated by the authors. This map is a schematic representation and may not reflect precise geographic accuracy.

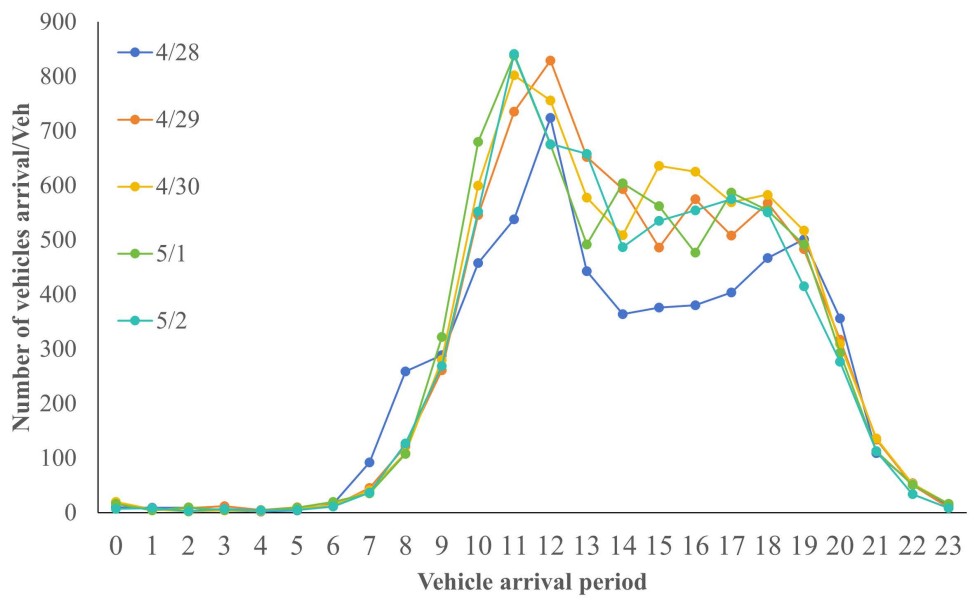

**Fig 2. Vehicle arrival timing on short holidays.**

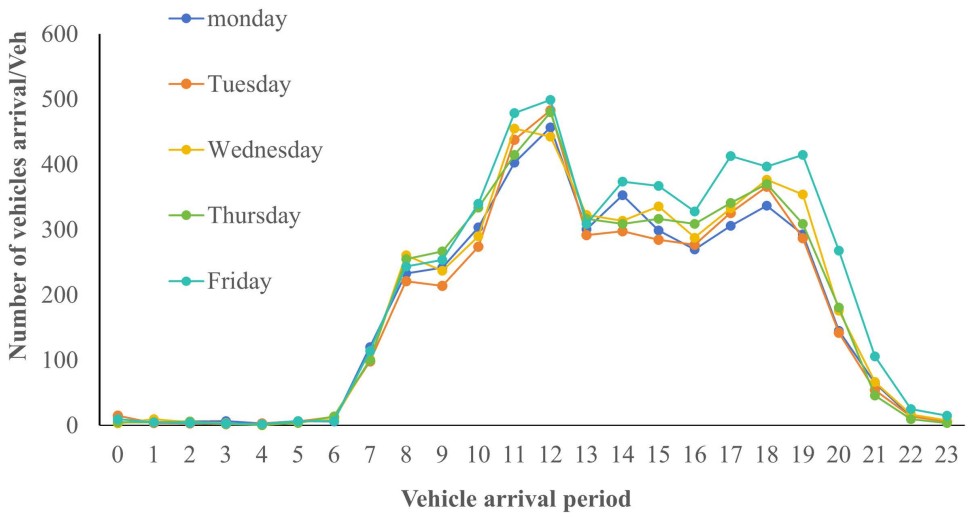

**Fig 3. Average Vehicle Arrival Time on Weekdays for a Month.**

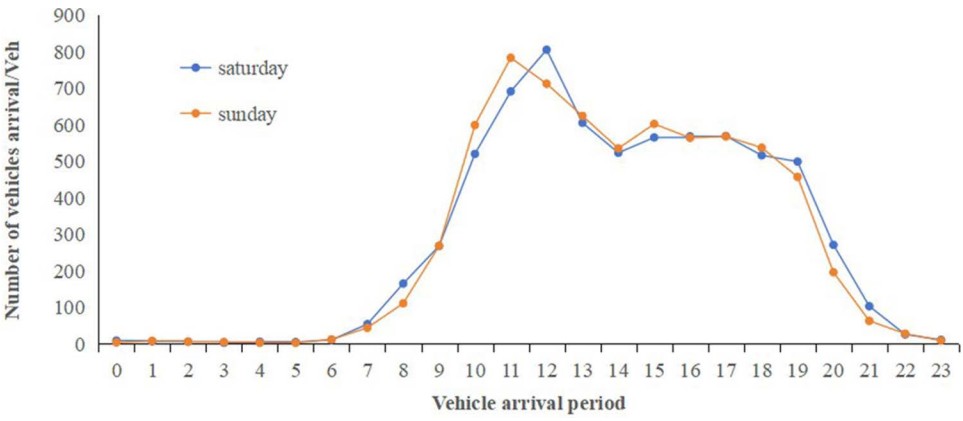

**Fig 4. Average Vehicle Arrival Time on Weekends for a Month.**

In summary, it can be observed that the parking space supply at this shopping center is generally sufficient. However, it can be seen from the peak parking saturation level that during the daytime on holidays, parking spaces are in high demand and highly utilized, while nighttime utilization is very low as the shopping center is closed at night. During weekdays, parking spaces are tight during peak periods.

## 4.2 Raw data processing

The study period is from 9:30 AM to 10:00 PM daily, where the need for parking is significant. Weather conditions that significantly affect parking activities were obtained using historical weather data. Parking fees were calculated based on the mall's pricing standards. The study distinguished between short holidays and regular weekends as separate variables. All variables and their attributes used to develop the survival model are provided in Table 2. A total of 166,836 records were compiled for this study. A summary of data breakdown by different categories is shown in Table 3. In the original dataset,

**Table 1. Average parking inventory over a 24-hour period each day.**

| Day char-acteristics | Daytime turn-over rate(times) | Nighttime turn-over rate(times) | Peak parking saturation (%) |
|---|---|---|---|
| Holiday | 4.2 | 0.28 | 100 |
| Weekday | 2.84 | 0.16 | 100 |

**Table 2. Variable attributes.**

| Variable | Symbol | Data type | Assignment |
|---|---|---|---|
| Day type | $X_1$ | Categorical | 1.short holidays; 2.weekday; 3.weekend |
| Weather | $X_2$ | Categorical | 1.Clear; 2.Light rain; 3.Moderate rain; 4.Heavy rain |
| Parking fee | $X_3$ | Numerical | Cost data/yuan |
| Entry time | $X_4$ | Numerical | Time data |
| Parking duration | $Y_1$ | Numerical | Duration data/h |
| Event status(whether departed) | $Y_2$ | Categorical | 1.Departed; 0.Censored |

**Table 3. Data sample example.**

| Number | Entry time | Parking fee/yuan | Weather | Data type | Parking duration/h | Event status |
|---|---|---|---|---|---|---|
| 1 | 19:56:45 | 5 | 1 | 1 | 0.98 | 1 |
| 2 | 20:20:07 | 7 | 1 | 1 | 1.08 | 1 |
| 3 | 18:16:09 | 9 | 1 | 1 | 2.65 | 1 |
| 4 | 14:39:42 | 21 | 1 | 1 | 8.15 | 0 |

there were 3 unlicensed vehicles, 15 police cars, and 10 military vehicles, which were excluded from the study. However, the entry and exit dates and times were retained to calculate parking duration.

## 5 Calculation analysis

### 5.1 Definitions of survival analysis

To convert parking-related data into a format suitable for survival analysis, the elements of the survival analysis problem are defined as follows:

(1) Survival Time: The period from when parking occurs to when it ends. This paper defines survival time as the period from entering the parking lot to leaving it within the observation window.

(2) Event Initiation: This reflects the initial characteristics of the subject's survival process. This paper sets it as the start of entering the parking lot within the observation window.

(3) Event Occurrence: Refers to the "death" event in survival analysis, that is, the occurrence of a specific outcome of interest. In this paper, the event is marked by the departure from the parking lot.

(4) Right censoring: Refers to the data for which the event has not been observed due to exceeding the observation time window. In this paper, vehicles not exited by 10:00 PM are considered censored data, specifically right-censored.

Statistical analyses were conducted using IBM SPSS Statistics v29, which provides robust computational capabilities for survival analysis through its dedicated survival analysis module. This software enabled efficient handling of complex survival models and facilitated the generation of graphical display of survival analysis results.

## 5.2 One-way survival analysis based on PL estimation

The PL estimation method was used to analyze the characteristics of parking duration under various types of days (week-days, weekends, and short holidays) and weather conditions (including no rain, light rain, moderate rain, and heavy rain). The model was calculated using maximum likelihood estimation. The results showed that the Log-Rank test P-values for different days and weather conditions were all less than 0.001, indicating statistical significance. The detailed analysis results are shown in Table 4, and the survival and hazard functions are shown in Figs 4 and 5.

Based on the table above, it is evident that the type of day (weekday, weekend, Short holidays) significantly affects parking duration. During Short holidays, the proportion of censored data reaches 9.3%, while weekdays have a censoring rate of 7.7% and weekends have 8.2%, which suggests that the probability of parking duration exceeding 10 PM is relatively higher during Short holidays. High parking demand after 10 PM during Short holidays is likely as shoppers are looking to spend late at night outside and not worry about going to work or school early morning the next day. Regarding the impact of

**Table 4. The PL estimation result.**

| Influencing factor | Variable | Total count | Event count | Censored | | Log-Rank Test |
|---|---|---|---|---|---|---|
| | | | | Case count | Percentage | |
| Day classification | Short holidays | 37662 | 34169 | 3493 | 9.3% | Chi2(2)=1140.036 $P < 0.001$ |
| | Weekdays | 84258 | 77738 | 6520 | 7.7% | |
| | Weekends | 44916 | 41217 | 3699 | 8.2% | |
| Weather | Clear | 99260 | 91222 | 8038 | 8.1% | Chi2(3)=127.604 $p < 0.001$ |
| | Light rain | 54007 | 49632 | 4375 | 8.1% | |
| | Moderate rain | 7021 | 6088 | 933 | 13.3% | |
| | Heavy rain | 6548 | 6182 | 366 | 5.6% | |

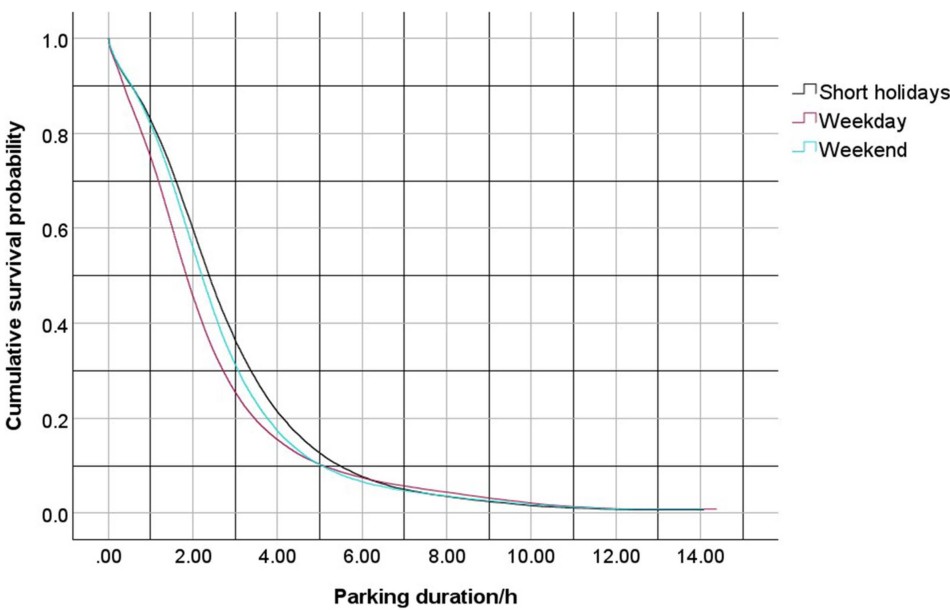

**Fig 5. Survival curve of parking duration on different days.**

weather on parking duration, the censoring rate accounts for 8.1% in clear weather, 8.1% in light rain, 13.3% in moderate rain, and 5.6% in heavy rain, which implies that the probability of parking duration exceeding 10 PM is very high in moderate rain. People may see driving during moderate as unsafe and try to stay inside the shopping mall longer for weather conditions to improve. The survival function distribution under the influence of different covariates is shown in Figs 5 and 6.

Based on Figs 5 and 6, the survival probability for parking duration declines significantly after approximately 2 hours, and this decline slows down after the duration reaches 3 hours. The survival functions for different days show that on weekdays, the survival probability is the lowest after 2 hours of parking, followed by weekends. After 3 hours of parking, the survival probability is highest during long holidays. Overall, the differences in survival probabilities among these three categories are not substantial. Regarding weather, the survival functions under no rain, light rain, and heavy rain show minimal differences. However, during moderate rain, the survival function is the highest. This shows that during moderate rain, individuals who travel for shopping are more likely to stay in the mall to wait for the rain to lessen before leaving, whereas, during heavy rain, individuals are more likely to leave the parking lot early and return home due to the unpredictability of the rain's intensity.

### 5.3 Multi-factor survival analysis based on cox semi-parametric approach

The analysis aims to investigate the impact of various factors on parking duration using the Cox proportional hazards model. Prior to the analysis, categorical variables among the covariates were converted into dummy variables, as shown in Tables 5 and 6.

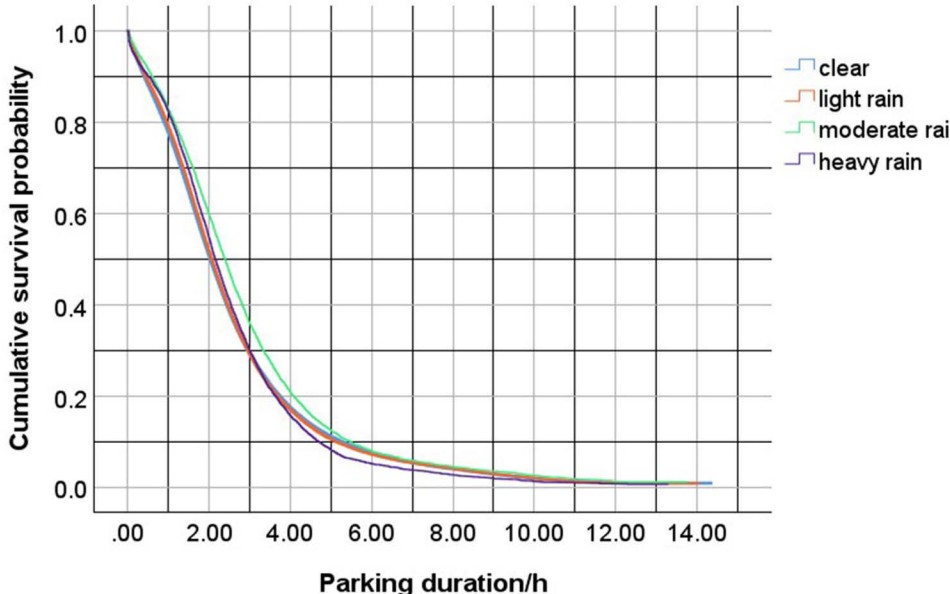

**Fig 6. Survival curve of parking duration in different weather.**

**Table 5. Day factor dummy variable.**

| Day classification | Dummy variable | |
| --- | --- | --- |
| | $X_{11}$ | $X_{12}$ |
| Short holidays | 0 | 0 |
| Weekday | 1 | 0 |
| Weekend | 0 | 1 |

The model test results yielded P < 0.05, indicating overall significance. Significant variables were retained based on the analysis results, as shown in Table 7.

In the multivariate analysis, the variables significantly influencing parking duration included weekday status, weekend status, moderate rain, and heavy rain. The regression coefficients for these variables were 0.068, 0.042, -0.089, and 0.030, respectively. Thus, the Cox regression function expression is:

$$\frac{h(t, X)}{h_0(t)} = \exp(0.068X_{11} + 0.042X_{12} - 0.089X_{22} + 0.030X_{23})$$

(9)

Interpreting the results reveals that the regression coefficient for the weekday factor is 0.068, with a relative risk ratio of 1.071. These findings demonstrate that weekday parking durations are statistically significantly shorter than those during short holidays, with a 7.1% increase in departure probability. The regression coefficient for the weekend variable is 0.042, with a relative risk ratio of 1.043. This suggests that weekends increase the probability of departing by 4.3% compared to short holidays. Regarding weather conditions, the regression coefficient for moderate rain is -0.089, with a relative risk ratio of 0.914. These results suggest that moderate rainfall is associated with prolonged parking durations compared to clear weather conditions, corresponding to an 8.6% reduction in departure probability. The regression coefficient for heavy rain is 0.030, with a relative risk ratio of 1.030. This indicates that heavy rain increases the probability of departing early compared to clear weather by 3%. Notably, the differences in survival curves are not substantial, which can be seen by low p-values. This result is also consistent with the previous univariate PL estimates. The survival function curve and cumulative hazard based on the mean values of the covariates are shown in Figs 7 and 8, respectively.

Fig 7 shows that most parking durations in the sample are concentrated within 2 hours, with a sharp decline in survival probability beyond 2 hours. Fig 8's cumulative hazard curve indicates that, under the influence of multiple factors, the probability of departing increases gradually after 2 hours, and parking duration in most cases does not exceed 3 hours.

**Table 6. Weather factor dummy variable.**

| Weather | Dummy variable | | |
|---|---|---|---|
| | $X_{21}$ | $X_{22}$ | $X_{23}$ |
| Clear | 0 | 0 | 0 |
| Light rain | 1 | 0 | 0 |
| Moderate rain | 0 | 1 | 0 |
| Heavy rain | 0 | 0 | 1 |

**Table 7. Cox regression analysis results.**

| Influencing factor | Explanatory variable | Regression coefficient | Hazard ratio | P-value |
|---|---|---|---|---|
| Day classification | $X_{11}$ | 0.068 | 1.071 | < 0.001 |
| | $X_{12}$ | 0.042 | 1.043 | < 0.001 |
| Weather | $X_{22}$ | -0.089 | 0.914 | < 0.001 |
| | $X_{23}$ | 0.030 | 1.030 | 0.034 |

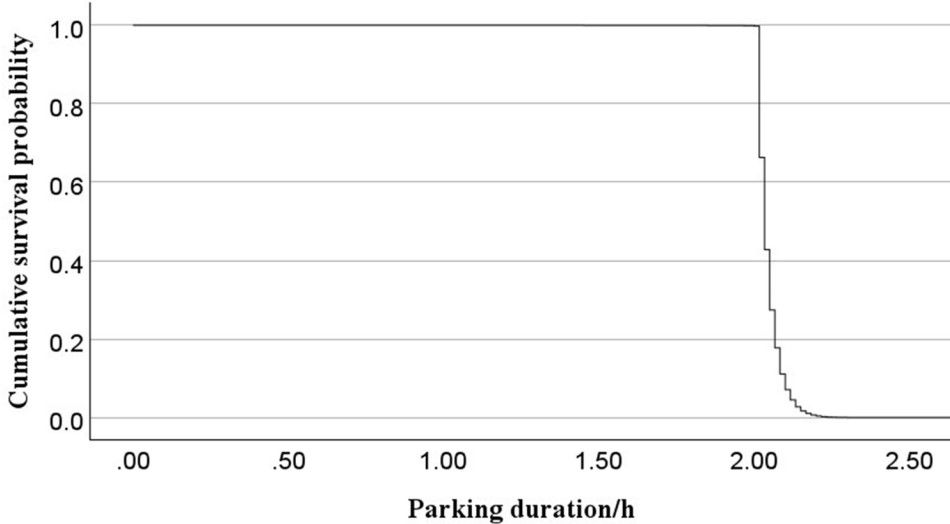

**Fig 7. Cox survival function curve.**

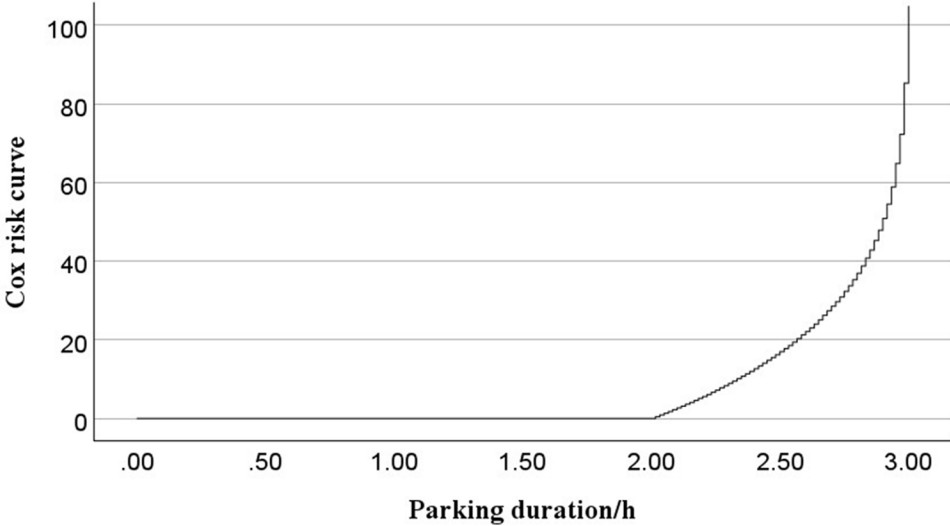

**Fig 8. Cox risk curve.**

## 6 Conclusions

A survival analysis model was developed to analyze the characteristics of parking duration in a commercial center's parking lot, with parking duration and departure status as the dependent variables. The model was applied to real parking data from a commercial center in Zhengzhou over 35 days, and the relevant parameters of the Cox proportional hazards model were estimated. A quantitative analysis of the factors influencing parking duration was conducted. The main conclusions are as follows:

(1) Most parking durations are concentrated within 2 hours. Beyond this duration, the probability of vehicles ending their parking gradually increases, with the longest duration extending up to 3 hours.

(2) Parking durations vary slightly under different days and weather conditions. In terms of data classification, compared to long holidays, whether it is a weekday or weekend has a slight impact on parking duration. Regarding weather conditions, compared to clear weather, moderate and heavy rain factors affect parking duration.

(3) Regression analysis revealed that parking fees did not significantly impact parking duration, which is contrary to common belief. The likely reason is that the research focused on vehicles entering underground paid parking lots, and once a paid parking lot is selected, there is a certain psychological expectation of the cost. Therefore, the impact on parking duration is insignificant, indicating that the current fee structure has room for improvement.

The research findings provide theoretical implications for the refined management of parking lots and the implementation of intelligent parking facilities. For example, the research results showed that different weather conditions and dates had varying effects on parking duration. Based on this, parking lot managers could adjust their management measures, such as implementing a flexible pricing system to increase or decrease parking fees according to time and weather changes and installing intelligent traffic information display devices to inform parkers in advance about weather changes, to ensure a more reasonable allocation of parking resources.

However, due to the data acquisition date limitations, only rainy weather was considered in this paper. Other weather conditions such as snow and heavy fog were not taken into account. Future research will expand the data to encompass an entire year to further refine the research results.

## Supporting information

**S1 Data. Desensitized original data.**

(7Z)

## Author contributions

**Conceptualization:** Zhendong Sun, Jiangling Wu, Feng Wang, Zhenzhong Tian.

**Data curation:** Zhendong Sun, Feng Wang.

**Formal analysis:** Zhendong Sun, Jiangling Wu.

**Methodology:** Zhendong Sun, Jiangling Wu.

**Validation:** Zhenzhong Tian.

**Visualization:** Zhendong Sun, Feng Wang.

**Writing – original draft:** Zhendong Sun, Jiangling Wu, Qiang He.

**Writing – review & editing:** Zhendong Sun, Jiangling Wu, Qiang He.

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
