## [Decision Letter · Decision Letter 0]

28 Nov 2024

PONE-D-24-25465Survival analysis of parking duration in urban commercial areas: A case study in Zhengzhou, ChinaPLOS ONE

Dear Dr. Wu,

Thank you for submitting your manuscript to PLOS ONE. After careful consideration, we feel that it has merit but does not fully meet PLOS ONE’s publication criteria as it currently stands. Therefore, we invite you to submit a revised version of the manuscript that addresses the points raised during the review process.

We look forward to receiving your revised manuscript.

Kind regards,

Yongxiang Zhang, Ph.D.

Academic Editor

PLOS ONE

Journal Requirements:

 This research is supported by the Science and Technology Project of Henan Province (No. 232102240019).  

This research is supported by the Science and Technology Project of Henan Province (No. 232102240019). The authors would like to express their sincere thanks to the anonymous reviewers for their helpful comments and valuable suggestions on this paper.

 This research is supported by the Science and Technology Project of Henan Province (No. 232102240019).

Reviewers' comments:

Reviewer's Responses to Questions

**Comments to the Author**

1. Is the manuscript technically sound, and do the data support the conclusions?

Reviewer #1: Yes

Reviewer #2: Yes

Reviewer #3: Yes

2. Has the statistical analysis been performed appropriately and rigorously? 

Reviewer #1: Yes

Reviewer #2: Yes

Reviewer #3: Yes

3. Have the authors made all data underlying the findings in their manuscript fully available?

Reviewer #1: Yes

Reviewer #2: Yes

Reviewer #3: Yes

4. Is the manuscript presented in an intelligible fashion and written in standard English?

Reviewer #1: Yes

Reviewer #2: No

Reviewer #3: Yes

5. Review Comments to the Author

Reviewer #1: This manuscript analyzed the parking duration in urvan commercial areas. Research has certain significance for understanding parking behaviors. However, there are still several comments for improvement.

1. The sentence in abstract "Based on Cox semi-parametric multi-factor analysis results, the probability of a longer vehicle duration is higher under heavy rain weather in the ." is not complete.

2.At the end of the Introduction, it is better to introduce the structure of the paper.

3. Provide a geographical map of the survey area.

4.Please explain the significance of the research and its theoretical or practical significance in the conclusion.

5. Please explain the tool software used for model solving and case analysis.

Applicability and advantages of the selected software and method.

Reviewer #2: 1. In the abstract, please fix your last four sentences. It is either missing a word or words or does not provide a complete idea.

2. Please also check this work. They provided empirical parking duration based on survey and presented parking fee calculation. Parking duration distribution has also been illustrated and can be used in the future. Cite if related.

Magsino, E.R., Arada, G.P. and Ramos, C.M.L., 2022. An evaluation of temporal-and spatial-based dynamic parking pricing for commercial establishments. IEEE Access, 10, pp.102724-102736.

3. In the introduction part, elaborate on your major contributions. I suggest place them in an enumerated format so that the readers can clearly see your contributions. Add a last paragraph that outlines the paper discussion.

4. In Section 2.2, what is PL? avoid acronyms in titles.

5. K-M estimate must be referenced in 2.2

6. Italicize your variables such as T, n, N, t_1, etc.

7. What do you mean by a "censored data"? What is a right-censored data? is there a left-censored data?

8. You have equation labels, (x), but they are not referenced in your text. Please include for easy referencing.

9. Enhance the readability of equation (5). Align your conditions.

10. In (6) is X defined as a linear combination of various X's? I did not see this assumption or definition.

11. Does the parking fee calculation in your study following fixed and linear rates defined in the suggested paper in comment 2? If so, Just place a formula for quick reference.

12. In Fig 1, what do you mean by "short holidays"? Which are short holidays in your dataset? In Figs. 2 and 3, are these the average numbers for the dataset? Please label your graphs correctly.

13. "On non-holidays" ==> During non-holidays

14. In Table 1, this should be average values, right?

15. In 3.1, you mentioned in your last sentence that parking spaces are tight during peak periods but this claim is not seen in your Table 1. Please include.

16. In Section 4.2, you mentioned "long holiday", but I don't see it in your Table 4. Confusing to read and understand.

17. Do you consider the curves in Figs 4 and 5 to have much difference? Please highlight if there is because from my point of view, there is not much difference that can be considered significant.

18. Figs 6 and 7 can be combined to provide clearer understanding.

Reviewer #3: The authors conducted analysis to investigate parking in urban commercial areas in Zhengzhou, China. The study investigates if and how weather, parking fees, and day of the week affect parking times and duration using non-parametric and semi-parametric survival analysis techniques. Some type errors should be corrected. The analysis, results, and conclusions are reasonably accurate. The literature is week and needs more recent references, for instance

Khedaywi, T., Al-Masaeid, H., Haddad, M., & Al-Ajlouni, S. (2023). VEHICLE PARKING AVAILABILITY IN THE CENTRAL BUSINESS DISTRICT OF IRBID CITY - JORDAN (A CASE STUDY). Journal of Engineering Science and Technology, 18(5), 2444-2469.

The reviewer recommends the paper for publication with minor corrections.

6. PLOS authors have the option to publish the peer review history of their article (what does this mean? ). If published, this will include your full peer review and any attached files.

**Do you want your identity to be public for this peer review?** For information about this choice, including consent withdrawal, please see our Privacy Policy .

Reviewer #1: **Yes: ** Yunqiang Xue

Reviewer #2: No

Reviewer #3: **Yes: ** Madhar Haddad

---

## [Author Response · Author response to Decision Letter 1]

15 Jan 2025

PONE-D-24-25465

“Survival analysis of parking duration in urban commercial areas: A case study in Zhengzhou, China” by Zhendong Sun, Jiangling Wu, Feng Wang, Zhenzhong Tian, Qiang He

PLOS ONE

Dear Editors and Reviewers,

We are submitting our revised manuscript titled "Survival Analysis of Parking Duration in Urban Commercial Areas: A Case Study in Zhengzhou, China" (PONE-D-24-25465) for consideration as an original research article in the journal PLOS ONE. We sincerely appreciate the time and effort invested by the editorial team at PLOS ONE in reviewing our manuscript, and we are grateful for the constructive feedback provided by the reviewers and the editor, which has been instrumental in enhancing the quality of our research.

In response to the reviewers’ comments, we have carefully revised the manuscript to address all comments and concerns. All modifications have been highlighted in the revised version using red text for clarity. Additionally, a detailed point-by-point response to the comments from reviewers is provided in “Responses (R) to Reviewers’ Comments (C)”.

This research was supported by the Science and Technology Project of Henan Province (No. 232102240019). As per our funding disclosure, "The funders had no role in study design, data collection and analysis, decision to publish, or preparation of the manuscript."

Regarding the issue of data sharing the editor mentioned, the experimental data is backend management data provided by the property management department of the target research area, under the coordination of the Fifth Detachment of Zhengzhou Traffic Police. The ownership of this data belongs to the property management department. Furthermore, the raw data contains potentially identifiable information, such as vehicle license plate details, and therefore cannot be publicly disclosed. However, interested parties may request access to the data by contacting szdszd321@hnp.edu.cn for further authorization.

We are confident that we have adequately addressed all the raised questions from the reviewers. Therefore, we sincerely hope that the revised manuscript receives the appropriate attention for potential publication in PLOS ONE.

Yours sincerely,

Dr. Jiangling Wu

(On behalf of all the authors)

School of Civil Engineering and Architecture, Henan University

Email: wujiangling_2006@126.com; wujiangling@henu.edu.cn

Responses (R) to Reviewers’ Comments (C)

First of all, thanks a lot for the reviewers’ advices. These comments are very valuable and helpful for improving our article. According to the reviewers’ comments, we have carefully modified our manuscript to meet the requirements of PLoS One.

Reviewer: 1

C (1): The sentence in abstract "Based on Cox semi-parametric multi-factor analysis results, the probability of a longer vehicle duration is higher under heavy rain weather in the." is not complete.

R (1): We are deeply embarrassed by this oversight, and the sentence has been completed in the manuscript. The details are as follows:

“Based on Cox semi-parametric multi-factor analysis results, the probability of a longer vehicle duration is higher under heavy rain weather in the study area.”

C (2): At the end of the Introduction, it is better to introduce the structure of the paper.

R (2): Thanks a lot for the Reviewer’s suggestion. They have been incorporated into the manuscript. The specific changes are as follows:

“The structure of this paper is as follows: In section 1, we introduced the concept and principles of survival analysis and analyzed the feasibility of its application in studying the characteristics of parking duration. In section 2, we constructed a PL non-parametric survival estimation model, and a Cox semi-parametric survival analysis model based on univariate and multivariate analyses, respectively. In section 3, we preprocessed the original parking data obtained and provided a descriptive analysis of the study area. In section 4, we input the processed data into the survival analysis models constructed in the second part for computation and discussed the results. In section 5, we summarized the entire experimental results and proposed corresponding suggestions for parking management based on the conclusions.”

C (3): Provide a geographical map of the survey area.

R (3): Thanks for the Reviewer’s comment. We have added the relevant figures to the section. 3.1 Overview of the study area. Fig.1 The diagram of study area

C (4): Please explain the significance of the research and its theoretical or practical significance in the conclusion.

R (4): Based on the reviewer's comments, we have updated the conclusion section:

“The research findings provide theoretical implications for the refined management of parking lots and the implementation of intelligent parking facilities. For example, the research results showed that different weather conditions and dates had varying effects on parking duration. Based on this, parking lot managers could adjust their management measures, such as implementing a flexible pricing system to increase or decrease parking fees according to time and weather changes and installing intelligent traffic information display devices to inform parkers in advance about weather changes, to ensure a more reasonable allocation of parking resources.”

C (5): Please explain the tool software used for model solving and case analysis.

Applicability and advantages of the selected software and method.

R (5): Thanks a lot for the Reviewer’s suggestion. The software used for model computation in our study is SPSS version 29, and this has been clarified in the revised manuscript:

“This study utilized SPSS version 29 software to perform calculations on the model. This software boasted efficient data processing capabilities and included a survival analysis calculation module, enabling it to compute complex survival analysis models and support the graphical display of survival analysis results.”

Reviewer: 2

C (1): In the abstract, please fix your last four sentences. It is either missing a word or words or does not provide a complete idea.

R (1): Thanks a lot for the Reviewer’s suggestion. We have carefully read the abstract section and revised the sentences.

C (2): Please also check this work. They provided empirical parking duration based on survey and presented parking fee calculation. Parking duration distribution has also been illustrated and can be used in the future. Cite if related.

Magsino, E.R., Arada, G.P. and Ramos, C.M.L., 2022. An evaluation of temporal-and spatial-based dynamic parking pricing for commercial establishments. IEEE Access, 10, pp.102724-102736.

R (2): Thanks a lot for the reference recommended by the Reviewer. We believe it is very inspiring for our current and future research, and we have already cited it in our article.

C (3): In the introduction part, elaborate on your major contributions. I suggest place them in an enumerated format so that the readers can clearly see your contributions. Add a last paragraph that outlines the paper discussion.

R (3): Thanks a lot for the Reviewer’s suggestion. We have made the revisions as suggested at the end of the introduction, and the changes are highlighted in red in the revised document.

C (4): In Section 2.2, what is PL? avoid acronyms in titles.

R (4): Thanks for the Reviewer’s comment. This error has been corrected in the revised version.

C (5): K-M estimate must be referenced in 2.2

R (5): Thanks for the Reviewer’s comment. This error has been corrected in the revised version.

C (6): Italicize your variables such as T, n, N, t_1, etc.

R (6): Thanks again for pointing out the errors in our article. They have been corrected in the revised version.

C (7): What do you mean by a "censored data"? What is a right-censored data? is there a left-censored data?

R (7): Censored data refers to instances where the event of interest (such as illness or death) does not occur within the observation window for some study subjects or where survival time cannot be observed due to reasons such as loss of follow-up or withdrawal from the study. Censoring includes left censoring, interval censoring, and right censoring. Right censoring refers to the situation where the event has not occurred by the end of the observation period but may occur at some point in the future. Left censoring refers to the situation where the event has already occurred before the observation period begins, but the exact time of occurrence is unknown. Interval censoring: refers to the situation where the event occurs during the observation period, but the exact time of occurrence is only known to be within a certain interval. The right-censored data in this paper refers to the data of vehicles that remain parked beyond the observation period. There is no left-censored data or interval-censored data in this paper.

C(8):You have equation labels, (x), but they are not referenced in your text. Please include for easy referencing.

R (8): Thanks a lot for the Reviewer's feedback. We have now included references to the equation labels (x) in the text as per your suggestion. This should make it easier for readers to locate and refer to the specific equations mentioned in the paper.

C (9): Enhance the readability of equation (5). Align your conditions.

R (9): Thanks a lot for the Reviewer’s suggestion. We have adjusted the format of Equation (5) to make it more readable.

C (10): In (6) is X defined as a linear combination of various X's? I did not see this assumption or definition.

R (10): Thanks a lot for the Reviewer careful review and the question regarding the definition of X in formula (6). We appreciate the opportunity to clarify this point. In the article, X is defined as a vector of k covariates, denoted as X = (X1, X2, …, Xk). These covariates represent the different independent variables that are believed to influence the parking duration of a vehicle in the context of the Cox regression analysis model. To clarify, X is not defined as a linear combination of various X's in this context. Rather, it is a collection or a set of individual covariates that are each considered separately in the model. The hazard function h (t, x) at time t under the influence of these covariates is then expressed mathematically in formula (6).

C (11): Does the parking fee calculation in your study following fixed and linear rates defined in the suggested paper in comment 2? If so, Just place a formula for quick reference.

R (11): Thanks very much for the Reviewer's suggestions. It should be clarified that the charging method in the target area of our study differs from that in the literature mentioned in Comment 2. However, we have followed the reviewer's advice and constructed a formula (7) for it for quick reference.

C (12): In Fig 1, what do you mean by "short holidays"? Which are short holidays in your dataset? In Figs. 2 and 3, are these the average numbers for the dataset? Please label your graphs correctly.

R (12): Thanks for the Reviewer’s comment. In China, "short holidays" refer to shorter vacations formed by adjusting weekend rest days to connect with statutory holidays, typically resulting in 3-day or 5-day breaks for the five festivals of New Year's Day, Tomb-Sweeping Day, International Workers' Day, Dragon Boat Festival, and Mid-Autumn Festival. For example, when a festival falls on a Saturday or Sunday, the holiday may be extended by one working day to form a continuous 3-day break; alternatively, by borrowing rest days from the previous or following week, a 3-day or 5-day vacation can be pieced together.

In our dataset, the data from April 28th to May 2nd, spanning five days, falls under the category of short holidays.

Figure 2 and Figure 3 indeed depict average values, and we have made the correction in the text.

C (13):"On non-holidays" ==> During non-holidays

R(13):Thanks for the Reviewer’s comment. This error has been corrected in the revised version.

C (14): In Table 1, this should be average values, right?

R(14): Thanks a lot for the Reviewer's feedback. Yes, Table 1 presents the average values, and we have updated the title of Table 1 to make it more precise.

C (15): In 3.1, you mentioned in your last sentence that parking spaces are tight during peak periods but this claim is not seen in your Table 1. Please include.

R�15��Thanks for the Reviewer’s comment. The conclusion regarding the shortage of parking spaces during peak hours is drawn from the value of average daily peak-hour parking lot saturation shown in Table 1.

C (16): In Section 4.2, you mentioned "long holiday", but I don't see it in your Table 4. Confusing to read and understand.

R (16): Thanks for the Reviewer’s comment. This is an expression error due to habit, and we have revised it to "short holiday."

C (17): Do you consider the curves in Figs 4 and 5 to have much difference? Please highlight if there is because from my point of view, there is not much difference that can be considered significant.

R (17): Thanks a lot for the Reviewer’s suggestion. Figure 4 and Figure 5 in the original manuscript describe the cumulative survival probability of parking duration under different dates and different weather conditions, respectively. Although they are distinct, the two curves for "clear" and "light rain" in Figure 5 exhibit very small differences and are almost overlapping, making them difficult to distinguish. In fact, there are four curves in Figure 5. We use highly contrasting colors to represent these two curves so that they can be more easily discernible.

C(18):Figs 6 and 7 can be combined to provide clearer understanding.

R(18): Thanks a lot for the Reviewer’s valuable suggestion to combine Figs 6 and 7 for a clearer understanding. However, these two figures are directly generated by the SPSS software, and due to the nature of the software's output, they cannot be directly merged into a single figure. We are open to your further guidance on the best approach to take, and we will ensure that the final version of the manuscript clearly presents the information in a way that enhances understanding.

Reviewer: 3

C(1): The authors conducted analysis to investigate parking in urban commercial areas in Zhengzhou, China. The study investigates if and how weather, parking fees, and day of the week affect parking times and duration using non-parametric and semi-parametric survival analysis techniques. Some type errors should be corrected. The analysis, results, and conclusions are reasonably accurate. The literature is week and needs more recent references, for instance

Khedaywi, T., Al-Masaeid, H., Haddad, M., & Al-Ajlouni, S. (2023). VEHICLE PARKING AVAILABILITY IN THE CENTRAL BUSINESS DISTRICT OF IRBID CITY - JORDAN (A CASE STUDY). Journal of Engineering Science and Technology, 18(5), 2444-2469.

R (1): Thanks very much for the Reviewer’s recognition and suggestions. The reference recommended by the reviewer is extremely valuable to us, and we have already cited it in the paper. In addition, we have made corrections to the errors in the paper according to the reviewer's suggestions and have also supplemented relevant references.

---

## [Decision Letter · Decision Letter 1]

26 Jan 2025

PONE-D-24-25465R1Survival analysis of parking duration in urban commercial areas: A case study in Zhengzhou, ChinaPLOS ONE

Dear Dr. Wu,

Thank you for submitting your manuscript to PLOS ONE. After careful consideration, we feel that it has merit but does not fully meet PLOS ONE’s publication criteria as it currently stands. Therefore, we invite you to submit a revised version of the manuscript that addresses the points raised during the review process.

We look forward to receiving your revised manuscript.

Kind regards,

Yongxiang Zhang, Ph.D.

Academic Editor

PLOS ONE

Journal Requirements:

Reviewers' comments:

Reviewer's Responses to Questions

**Comments to the Author**

1. If the authors have adequately addressed your comments raised in a previous round of review and you feel that this manuscript is now acceptable for publication, you may indicate that here to bypass the “Comments to the Author” section, enter your conflict of interest statement in the “Confidential to Editor” section, and submit your "Accept" recommendation.

Reviewer #1: All comments have been addressed

Reviewer #2: All comments have been addressed

Reviewer #3: All comments have been addressed

2. Is the manuscript technically sound, and do the data support the conclusions?

Reviewer #1: Yes

Reviewer #2: Yes

Reviewer #3: Yes

3. Has the statistical analysis been performed appropriately and rigorously? 

Reviewer #1: Yes

Reviewer #2: Yes

Reviewer #3: Yes

4. Have the authors made all data underlying the findings in their manuscript fully available?

Reviewer #1: Yes

Reviewer #2: Yes

Reviewer #3: Yes

5. Is the manuscript presented in an intelligible fashion and written in standard English?

Reviewer #1: Yes

Reviewer #2: Yes

Reviewer #3: Yes

6. Review Comments to the Author

Reviewer #1: The manuscript has improved a lot. It is better to add quantitavie research results in the abstract. The authors should do a thorough proofreading of the entire manuscript before publication.

Reviewer #2: The authors have fully addressed all my comments and questions. They have revised and improved the previously submitted manuscript.

Reviewer #3: The authors have addressed the comments of the reviewer. The reviewer recommeds the paper for publication.

7. PLOS authors have the option to publish the peer review history of their article (what does this mean? ). If published, this will include your full peer review and any attached files.

**Do you want your identity to be public for this peer review?** For information about this choice, including consent withdrawal, please see our Privacy Policy .

Reviewer #1: No

Reviewer #2: No

Reviewer #3: **Yes: ** Madhar Haddad

---

## [Author Response · Author response to Decision Letter 2]

18 Mar 2025

Responses (R) to Reviewers’ Comments (C)

First of all, thanks a lot for the reviewers’ advices. These comments are very valuable and helpful for improving our article. According to the reviewers’ comments, we have carefully modified our manuscript to meet the requirements of PLoS One.

Reviewer: 1

C: The manuscript has improved a lot. It is better to add quantitavie research results in the abstract. The authors should do a thorough proofreading of the entire manuscript before publication.

R:Thanks a lot for the reviewers' affirmation and suggestions on our paper.We highly agree with the reviewer's suggestion to include quantitative research results in the abstract and have accordingly supplemented the abstract with relevant quantitative research findings to enhance its informativeness and appeal. The relevant portion of the revised abstract is:”the parking duration is significantly influenced by weekdays (regression coefficient = 0.068, hazard ratio=1.071, P < 0.001), weekends (regression coefficient = 0.042, hazard ratio=1.043,P < 0.001), moderate rain (regression coefficient = -0.089, hazard ratio=0.914, P < 0.001), and heavy rain (regression coefficient = 0.030, hazard ratio=1.030,P = 0.034 < 0.05). The results have indicated that within the study area, compared to short holidays, the parking duration on weekdays and weekends is shorter, with the probability of vehicles ending their parking increased by 7.1% and 4.3%, respectively. Under different weather conditions, compared to sunny days, parking duration is longer during moderate rain, with the probability of vehicles departing decreased by 8.6%, whereas during heavy rain, parking duration is shorter, with the probability of vehicles departing increased by 3%.”

Furthermore, we have thoroughly proofread the entire manuscript based on the suggestions, ensuring that all potential errors and inconsistencies are eliminated prior to publication. These revisions are highlighted in red font within the manuscript.

Reviewer: 2

C:The authors have fully addressed all my comments and questions. They have revised and improved the previously submitted manuscript.

R:The reviewer has indicated that we have fully addressed all their comments and questions, and have revised and improved the previously submitted manuscript. We greatly appreciate the reviewer's affirmation and encouragement, which has provided us with great motivation to continue refining the paper. We will continue to polish and optimize the paper based on the reviewer's suggestions.

Reviewer: 3

C:The authors have addressed the comments of the reviewer. The reviewer recommends the paper for publication.

R:We feel deeply honored to have received the recognition and support from the reviewer. We will continue to strive to enhance the quality of our paper and look forward to successfully publishing our research findings in this esteemed journal.

---

## [Decision Letter · Decision Letter 2]

4 Apr 2025

Survival analysis of parking duration in urban commercial areas: A case study in Zhengzhou, China

PONE-D-24-25465R2

Dear Dr. Wu,

We’re pleased to inform you that your manuscript has been judged scientifically suitable for publication and will be formally accepted for publication once it meets all outstanding technical requirements.

Kind regards,

Yongxiang Zhang, Ph.D.

Academic Editor

PLOS ONE

Additional Editor Comments (optional):

Reviewers' comments:

Reviewer's Responses to Questions

**Comments to the Author**

1. If the authors have adequately addressed your comments raised in a previous round of review and you feel that this manuscript is now acceptable for publication, you may indicate that here to bypass the “Comments to the Author” section, enter your conflict of interest statement in the “Confidential to Editor” section, and submit your "Accept" recommendation.

Reviewer #1: All comments have been addressed

2. Is the manuscript technically sound, and do the data support the conclusions?

Reviewer #1: Yes

3. Has the statistical analysis been performed appropriately and rigorously? 

Reviewer #1: Yes

4. Have the authors made all data underlying the findings in their manuscript fully available?

Reviewer #1: Yes

5. Is the manuscript presented in an intelligible fashion and written in standard English?

Reviewer #1: Yes

6. Review Comments to the Author

Reviewer #1: All the comments have been addressed. I am satisfied with the revised manuscript. The authors should do a thorough proofreading of the entire manuscript before publication.

7. PLOS authors have the option to publish the peer review history of their article (what does this mean? ). If published, this will include your full peer review and any attached files.

**Do you want your identity to be public for this peer review?** For information about this choice, including consent withdrawal, please see our Privacy Policy .

Reviewer #1: No

---

## [Editor Report · Acceptance letter]

PONE-D-24-25465R2

PLOS ONE

Dear Dr. Wu,

I'm pleased to inform you that your manuscript has been deemed suitable for publication in PLOS ONE. Congratulations! Your manuscript is now being handed over to our production team.

Kind regards,

on behalf of

Dr. Yongxiang Zhang

Academic Editor

PLOS ONE